# Association Between Dairy Production System and Milk Functionality Based on Analysis of miRNAs in Exosomes from Milk

**DOI:** 10.3390/ani14202960

**Published:** 2024-10-14

**Authors:** Loubna Abou el qassim, Regina Golan-Gerstl, Shimon Reif, Luis J. Royo

**Affiliations:** 1Department of Animal Nutrition, Servicio Regional de Investigación y Desarrollo Agroalimentario (SERIDA), 33300 Villaviciosa, Spain; royoluis@uniovi.es; 2Dairy Research & Innovation Centre, Scotland’s Rural College (SRUC), Barony, Parkgate, Dumfries DG1 3NE, UK; 3Pediatric Department, Hadassah Medical Center, Jerusalem 91120, Israel; reginag@hadassah.org.il (R.G.-G.); shimon@hadassah.org.il (S.R.); 4Department of Functional Biology, University of Oviedo, 33006 Oviedo, Spain

**Keywords:** dairy production systems, cow milk, extracellular vesicles, miRNA, milk bioactive potential

## Abstract

**Simple Summary:**

This study compares milk from extensive (pasture-based) and intensive (confined housing) dairy farms to see how different farming methods affect the miRNAs in milk exosomes. We found that milk from extensive farms had more numerous extracellular vesicles. Specifically, the miRNA *bta-miR-451* was found in higher levels in milk from extensive farms. This miRNA is similar to a human miRNA linked to important health pathways, including those related to Parkinson’s disease and cancer. These results suggest that milk from extensive systems may have better functional properties. Further research is needed to explore the potential health benefits of these miRNAs.

**Abstract:**

Dairy farming practices significantly affect the nutritional and functional properties of milk. This study compares miRNAs in milk exosomes from extensive and intensive dairy systems and explores their potential implications for human consumers. Extensive systems are believed to produce milk of higher quality with better animal welfare compared to intensive systems. Milk samples from eight extensive and nine intensive dairy farms were analysed. Milk-derived extracellular vesicles were isolated through sequential ultracentrifugation and characterised through Dynamic Light Scattering and Nanosight to determine the size and the concentration of the extracellular vesicles, in addition to immunoblotting to ensure the presence of exosome-specific proteins in their membrane. miRNA levels were quantified using RT-qPCR, and metabolic pathways associated with miRNAs showing significant differences between farm groups were analysed. EVs from extensive farms had higher concentrations. Notably, *bta-miR-451* levels were significantly higher in milk from extensive farms (*p* = 0.021). Like human miRNA *hsa-miR-451*, it is linked to pathways related to Parkinson’s disease and cancer. Our research suggests that milk production in extensive systems not only provides socioeconomic and environmental benefits but may also have positive effects on human health. Further research is warranted to explore the bioactive potential of these miRNAs and their implications for human health.

## 1. Introduction

Dairy farms can differ substantially in how they feed, house, milk, and care for cows [1,2]. In “extensive” dairy production systems, animals are mostly fed on pastures through grazing, taking advantage of the natural resources of the territory, with a low use of external inputs and a minimum supply of concentrates. This provides a lower feeding system cost whose efficiency is based on high milk production per unit area. In general, this system is characterised using breeds adapted to the territory and proper management of the reproduction and genetic diversity of the herd [3]. “Intensive” production systems, in contrast, involve housing cows within closed facilities and with a high stocking density (around 4.75 animal/ha) and the maximization of milk production per cow per unit area. Feeding is based on the use of silages and an increased use of concentrates per cow which makes these farms dependent on external feeding [1,3], feeding them primarily corn silage. Extensive production systems are generally believed to provide milk of higher quality while ensuring better animal welfare and environmental sustainability than intensive systems [3,4].

The type of milk production system affects the composition and levels of microRNAs (miRNA) in the milk [5,6], probably reflecting the fact that levels of miRNAs in milk are affected by numerous environmental factors, including diet [7], exercise [8], and stress [9]. These variations may be useful as authentication biomarkers, helping to distinguish between milk produced in extensive versus intensive systems [5,6]. As short, non-coding RNA molecules typically 20–24 nucleotides long, miRNAs play a key role in regulating gene expression by binding to complementary sequences on messenger RNA (mRNA) molecules, leading to their degradation or the inhibition of their translation into proteins [10].

Most miRNAs are found in the cellular environment, but some are present in body fluids, known as extracellular or circulating miRNAs [11]. 

These extracellular miRNAs, particularly those carried by exosomes [11], can be transferred to other cells, thanks to the mentioned vesicles enabling cell-to-cell communications [12,13]. In light of this, some studies challenge the traditional view of miRNAs as solely endogenous regulators, suggesting that dietary miRNAs such as those in milk can be absorbed by the human consumer, becoming bioavailable and potentially influencing biological processes beyond their origin, further suggesting the potential role to regulate gene expression in host cells across different species [14,15,16,17] and thus imparting different functional properties to the milk. Studies have shown that in cellular models, miRNAs contained within extracellular vesicles in milk can modify gene expression in recipient cells that internalize these vesicles [15,16,17], highlighting their potential functional role in human health.

To investigate the potential functions of these miRNAs and the metabolic pathways in which they are involved, various computational methods have been developed for predicting miRNA target genes. These methods rely on assessing the degree of sequence complementarity between a miRNA and its target [18]. Various tools and databases have been created, including DIANA-miRPath [19], TargetScan [20], and databases specific to ruminants, such as RumimiR [21]. Some tools, like TarBase, compile experimentally supported targets across different species, with information gathered manually from the literature [22]. However, perfect seed pairing is not always a reliable indicator of miRNA interactions, which may explain why some predicted target sites do not function as expected. Therefore, experimental verification of computationally identified targets is crucial [23].

In this study, we analysed and compared the miRNA profiles in exosomes from milk produced through extensive and intensive farming systems. Using bioinformatics, we explored how these differences could influence gene regulation in humans consuming the milk, potentially shedding light on the health implications of different production systems.

## 2. Materials and Methods

### 2.1. Isolation of Milk-Derived Extracellular Vesicles (MDEVs)

Milk was sampled from tanks on eight extensive and nine intensive dairy farms in Asturias, Spain between June and August 2021 (Table 1). The samples were stored at 4 °C and immediately transported to the laboratory, the milk was mixed thoroughly to ensure homogeneity, and 100 mL aliquots were taken for processing.

To begin the preparation process, each milk sample was centrifuged at 3000× *g* for 15 min at 4 °C. This step was performed to remove milk fat globules, somatic cells, and any cellular debris that could interfere with subsequent analyses. After centrifugation, the milk whey was transferred into fresh RNase-free centrifuge tubes. Next, 1% (*w*/*v*) acetic acid was added to the whey at a ratio of 300 µL per 40 mL of whey. The mixture was vortexed to ensure thorough mixing and then centrifuged again at 3000× *g* for 10 min. The casein formed a pellet, and the clear whey supernatant was filtered, collected, and stored at −80 °C for further processing.

The next stage of the procedure involved isolating extracellular vesicles from the whey. Sequential ultracentrifugation was employed for this purpose [24]. The whey was placed into OptiSeal ultracentrifuge tubes (14 mL, 95 mm; Beckman Coulter, Gladesville, Australia) and subjected to a series of centrifugation steps in a SW 40 Ti rotor (Beckman Coulter) at increasing speeds: 12,000× *g* for 60 min, 35,000× *g* for 60 min, and finally 70,000× *g* for another 60 min, all at 4 °C. To ensure purity, the final supernatant was passed through two syringe filters: first through a 0.45 μm filter, followed by a finer 0.22 μm filter. The filtered supernatant was then ultracentrifuged at 135,000× *g* for 90 min at 4 °C. This final high-speed centrifugation step pelleted the extracellular vesicles, which were resuspended in 300 μL of phosphate-buffered saline (PBS).

The resuspended pellet was allowed to sit at 4 °C for 12 h to ensure the vesicles were fully solubilized. After gently mixing the sample with a micropipette to break up any potential clumps, the extracellular vesicles were stored at −80 °C until further analysis.

These vesicles were characterised for their size, concentration, specific protein, and purity before RNA extraction for downstream analysis of miRNAs. 

### 2.2. Characterisation of MDEVs

#### 2.2.1. Size and Concentration

Vesicles were characterised through Dynamic Light Scattering (DLS) by 10-fold dilution in PBS, transfer into a ZEN 0040 disposable cuvette (Malvern Panalytical, Malvern, UK), and analysis on a Zetasizer (Malvern Instruments, Malvern, UK) equipped with a solid-state He-Ne laser emitting at 633 nm. Analyses were performed at 20 °C and a detection angle of 173° backscatter. Vesicles were also analysed using a NanoSight NS300 nanoparticle analyser (Malvern), which was used to analyse the size, concentration, and dynamic behaviour of MDEVs. Samples from the two groups of farms were pooled, and for each group, three serial concentrations were prepared: 1/100, 1/1000, and 1/10,000.

#### 2.2.2. MDEVs Specific Protein Determination

Total protein in vesicles was estimated using a bicinchoninic acid assay (BCA Pierce, Thermo Scientific, Waltham, MA, USA). An aliquot of vesicle suspension (50 µL) was mixed with an equal volume of 2 × RIPA buffer (Abcam, Cambridge, MA, USA) and 1 µL of protease inhibitor (cOmplete™, Roche, Basel, Switzerland), vigorously vortexed, and incubated on ice for 10 min. Duplicate aliquots (5 µL) were mixed with 200 µL bicinchoninic acid assay buffer (Thermo Scientific) and processed according to the manufacturer’s instructions, including 30 min incubation at 37 °C and determination of absorbance at 562 nm on a spectrophotometer (Tecan, Crailsheim, Germany). Total protein concentrations were determined based on a third-order polynomial calibration curve generated from duplicate samples containing known concentrations of bovine serum albumin. 

Exosome-specific proteins in the membrane of extracellular vesicles were detected using antibodies against CD 9 and CD 81 (Abcam, Cambridge, UK). Absence of casein in the membrane was confirmed using an antibody against the protein (Abcam, Cambridge, UK). An aliquot of vesicles in RIPA buffer (90 µL) was mixed with 30 µL 4× LSD buffer incubated at 70 °C for 10 min and fractionated by SDS-PAGE. Proteins were transferred to polyvinylidene difluoride membranes, which were incubated with primary antibody overnight at 4 °C, followed by horseradish peroxidase-conjugated secondary antibody (1:5000; Zymed Laboratories, San Francisco, CA, USA). Antibody binding was detected using enhanced chemiluminescence.

### 2.3. Analysis of miRNAs in MDEVs 

#### 2.3.1. miRNAs Extraction

Total RNA was extracted from vesicles using Trizol (INVITROGEN, Carlsbad, CA, USA) and the Zymo Direct-zol RNA MiniPrep Kit (Zymo Research, Irvine, CA, USA) as described [25]. 

The vesicles were lysed using TRIzol reagent and centrifuged at 12,000× *g* for 1 min at 4 °C. The resulting supernatant was treated with DNase I and incubated at room temperature for 15 min. This solution was then transferred to Zymo-Spin™ IC (Zymo Research, Irvine, CA, USA) columns and centrifuged again at 12,000× *g* for 1 min at 4 °C. The RNA was washed with RNA wash buffer and eluted with 40 μL of RNase-free water. 

The purity of the extracted RNA was assessed and quantified using the Nano-Drop 2000 (Thermo Scientific™, Wilmington, DE, USA).

Complementary DNA (cDNA) was synthesised from the extracted RNA using the High-Capacity RNA-to-cDNA Kit (Applied Biosystems, Foster City, CA, USA), with 1 mg of total RNA isolated from exosomes, following the procedure outlined in previous studies [16,25].

#### 2.3.2. Quantitative Reverse-Transcription Polymerase Chain Reaction (RT-qPCR)

Levels of the following four miRNAs were analysed in samples from extensive and intensive dairy farms using quantitative RT-qPCR involving a two-step cycling protocol as described [16,25]. Levels of the following four miRNAs were analysed in samples from extensive and intensive dairy farms using RT-qPCR involving a two-step cycling protocol as described [16,25,26], fast SYYBR Green Master Mix and the StepOnePlus Real-Time PCR System (Applied Biosystems, Foster City, CA, USA): *miR-148a*, levels of which in milk have been associated with feeding and metabolism [27,28]; *bta-miR-215*, high levels of which have been reported in milk from intensive farms [5]; *bta-miR-451*, high levels of which have been reported in milk from extensive farms [28,29]; and *bta-miR-30a-5p*, levels of which do not appear to differ between extensive or intensive farms [5,6]. Levels of each of these miRNAs were determined relative to that of the control RNA, RNU6. 

### 2.4. Prediction of Genes Targeted by miRNAs from MDEVs

Human genes whose mRNAs might be targeted by miRNAs whose levels differed significantly between milk samples from extensive or intensive dairy farms were predicted by identifying endogenous human miRNAs similar to the miRNAs from cows within the miRBase platform (version 22.1; http://www.mirbase.org, accessed on 20 June 2022). Putative mRNA targets of the human miRNAs were predicted using DIANA tools in TarBase (version 8), and pathways involving the human miRNAs were predicted using DIANA tools in miRPath (version 2.0) [19].

### 2.5. Statistical Analysis

Differences in miRNA levels between milk samples from extensive or intensive dairy farms were assessed for significance using the Kruskal–Wallis test within SPSS 22.0 (IBM, Chicago, IL, USA). Differences associated with *p* < 0.05 were considered statistically significant.

## 3. Results

### 3.1. Confirmation and Characterisation of Exosomes in Milk Samples

DLS indicated that the majority of extracellular vesicles in the 17 milk samples had diameters between 31.69 and 92.01 nm (average, 76.34 ± 34.76 nm) (Figure 1). The Z-value of the samples ranged from 141.10 to 184.70 nm (average, 160.15 nm), while the distribution polydispersity index ranged from 0.22 to 0.40 (average, 0.28). These analyses suggested that the extracellular vesicles isolated through our procedure were primarily exosomes, which are membrane-bound vesicles with diameters of 30–120 nm that are derived from endosomes [11]; the samples showed minimal contamination by microvesicles, which have diameters of 100–1000 nm [11].

Nanoparticle tracking analysis (NTA) indicated an average vesicle diameter of 166 nm from intensive dairy farms and 178 nm from extensive dairy farms (Figure 2), consistent with the results of DLS. The total concentration of vesicles was higher in milk from extensive farms (1.4 ± 0.2 × 10^9^ particles/mL) than in milk from intensive farms (6.8 ± 1.1 × 10^8^ particles/mL). Additionally, particle visualisation through NTA allowed for real-time observation of vesicles in suspension, enabling confirmation of their presence and movement via Brownian motion (Appendix A).

Following protein quantification, immunodetection with specific antibodies against CD 81 and CD 9 revealed bands corresponding to these proteins in the 17–20 kDa range, as shown in Figure 3. Additionally, the absence of casein, which shows a band in the 25–35 kDa range, was confirmed using casein-specific antibodies, thereby verifying the purity of the exosomes and ruling out the potential co-purification with casein micelles. 

### 3.2. Relative Levels of miRNAs in MDEVs

In line with these findings, we observed that *bta-miR-451* was significantly more abundant in MDEVs from extensive dairy farms than in vesicles in milk from intensive farms (*p* = 0.021), whereas the levels of the other miRNAs did not differ significantly between the two production systems (Figure 4). Nevertheless, levels of *bta-miR-148a* tended to be lower in milk from extensive farms (*p* = 0.114).

The miRNA *bta-miR-451* (sequence: aaaccguuaccauuacugaguuu) exhibits high homology with the human miRNA *hsa-miR-451a* (sequence: aaaccguuaccauuacugaguu). Assuming that bta-*miR-451* targets the same genes as its human counterpart, analysis with Tarbase revealed that 27 human target genes for *miR-451* have been experimentally validated, which are linked to four KEGG pathways (Table 2). These pathways are related to Parkinson’s disease, the mTOR signalling pathway, thyroid cancer, and colorectal cancer, suggesting a potential connection between extensive dairy farm management and enhanced functional properties of the milk produced. 

## 4. Discussion

Our work confirms the idea that the type of dairy production system influences the functional properties of milk. In our study, we provide evidence that an extensive dairy production system leads to milk containing more abundant extracellular vesicles and higher levels of the vesicle-associated miRNA *bta-miR-451*. This miRNA may be associated with Parkinson’s disease, the mTOR signalling pathway, thyroid cancer, and colorectal cancer, suggesting greater functional value in milk from extensive than intensive dairy farms. Our work confirms the idea that the type of dairy production system influences the functional properties of milk [6].

The pellets of EVs obtained after ultracentrifugation may contain exosomes, as well as other vesicles, macromolecules, and protein aggregates [24]. To ensure the proper identification of the isolated molecules, a series of analyses have been performed, as it is difficult to establish a clear classification of EVs due to their considerable variability and, most notably, their overlapping sizes. The term ‘extracellular vesicles’ was introduced to describe all vesicles secreted into biological fluids, whereas the use of the term ‘exosomes’ implies that certain criteria must be met, including their size (30–150 nm), their origin from endosomal multivesicular bodies, and the presence of specific protein markers (CD 9, CD 63, and CD 81). Additionally, exosomes should be isolated from extracellular fluids without cellular disruption, and their biophysical properties should be confirmed [30,31].

DLS analysis confirms that the most numerous particle population had diameters ranging from 31.69 nm to 92.01 nm, which falls within the recognised size range of exosomes. However, the average particle size varied between 141.10 nm and 184.70 nm, indicating that other EVs were also isolated. Vesicles with diameters of 80 to 300 nm correspond to the microsomal fraction, which includes exosomal particles [32], reflecting the complexity of separating these two groups using current techniques [24].

After confirming the presence of exosome-sized vesicles, the next step to verify that the isolated vesicles are exosomes is to ensure they carry specific markers, such as CD 9 and CD 81, using immunoblotting. Caseins are not intrinsic components of exosomes but may appear as co-isolated impurities [30], so immunoblotting was also used to confirm the purity of the samples and the absence of casein. Exosomal markers were clearly observed, as they were enriched in the exosomal fractions, while aggregated proteins were eluted in most samples. 

The concentrations of vesicles in milk from both types of farms in our study fell within the broad range in the literature (0.0048–7.2 × 10^11^ particles/mL) [24,33]. The large variation in vesicle concentrations can be attributed to differences in extraction methods [32] and physiological states [34]. Mitchell et al. (2016) [35] found that plasma exosome concentrations were 50% higher in fertile cows compared to those in subfertile ones. Dietary factors also affect milk exosome levels; replacing alfalfa hay and corn silage with whole cottonseed and soybean hulls increased exosome concentrations in cows fed these ingredients [36]. Changes in rumen fermentation, improved feed conversion rates, and high mineral content, particularly phosphorus, which is crucial for sphingolipid synthesis and EV biogenesis, further influence these concentrations [37].

We also assessed the levels of the studied miRNAs from exosomes in milk from intensive and extensive farms. One notable finding was the very low concentration of total RNA, which averaged around 4.85 ng/μL. Previous studies have reported RNA concentrations in bovine milk whey-derived exosomes ranging from 0.50 to 1 ng/μL [36]. However, another study showed higher results of 4.80 to 33 ng/μL [38]. This indicates that while the total RNA concentration in our study is relatively low, it falls within the range reported by some studies, indicating that our results are consistent with the existing literature.

After completing exosome characterisation, we analysed the exosomal miRNA, as they can influence recipient cell behaviour [12,13,14]. Levels of *bta-miR-451* were significantly higher in exosomes in milk from extensive farms than in milk from intensive farms, which agrees with our previous study [29] involving different cow feeding management in controlled conditions. Similarly, levels of *bta-miR-451* in exosomes from plasma or the biceps femoris muscle of Japanese Shorthorn cattle were significantly higher in grazing animals than in non-grazing animals [28], implying a relationship between the secretion of *bta-miR-451* by skeletal muscles and blood circulation during grazing. This leads us to suggest that the exercise through grazing may also increase levels of *bta-miR-451* in milk, especially given that some miRNAs in milk may originate from blood [39]. 

Whatever its cause(s), the observed higher content of *bta-miR-451* may improve the functional properties of milk, given that the human homologue *hsa-miR-451* has been proposed to regulate signalling pathways involved in Parkinson’s and other neurodegenerative diseases [40] and to suppress growth of, or promote apoptosis in, various types of cancer [41,42,43,44].

Notably, *hsa-miR-451* expression was significantly lower in Parkinson’s patients compared to that in healthy individuals, suggesting that reduced levels of this miRNA may contribute to post-transcriptional repression of mRNA targets associated with neurodegenerative pathways, including Parkinson’s disease [40]. In addition, *miR-451a* has been shown to act as a potential therapeutic agent in cancer treatment by targeting the tuberous sclerosis 1 (TSC1) gene, which activates the PI3K/Akt/mTOR signalling pathway in cancer cells. This miRNA has also been found to enhance the effects of anti-myeloma drugs by promoting cell apoptosis, reducing clonogenicity, and decreasing MDR1 mRNA expression [41]. Furthermore, *miR-451a* functions as a tumour suppressor in papillary thyroid carcinoma [42] and colorectal cancer by targeting the macrophage migration inhibitory factor (MIF) [43], helping to repress cell proliferation and invasion [44].

We focused on bulk tank milk samples rather than examining individual cow samples. While this approach provides a broader perspective of the overall production system, it averages out individual factors such as lactation stage, age, and specific cow health status, which could influence miRNA profiles in exosomes. The sampled milk was sourced from commercial farms that regularly conduct health screenings for somatic cell counts, ensuring that the included cows were healthy.

This study has several limitations that should be acknowledged. First, we did not investigate the influence of the chemical composition of the meal on miRNA levels in milk exosomes. Nutritional components can greatly affect milk characteristics and, consequently, miRNA profiles. Although previous research yielded inconclusive results regarding fat miRNAs (non-published), this area remains worthy of exploration in future studies.

Moreover, we did not explore the factor or factor combination in the production system behind the expression of miR-451; we related it to grazing, especially the exercise (not the fresh grass feeding), based on the study of Muroya et al., 2015 and De La Torre-Santos 2021 [28,29], but we still need to explore it in a controlled assay. Future research should address these gaps to provide a more comprehensive understanding of the factors influencing miRNA levels in milk exosomes.

Our findings suggest that milk from extensive farming systems, which contains higher levels of *miR-451a*, may have bioactive potential to affect human cells. Further studies should test this hypothesis through in vitro or in vivo assays to confirm the bioinformatic predictions.

## 5. Conclusions

In conclusion, our findings tentatively suggest potential differences in MDE concentrations and specific miRNA levels between extensive and intensive dairy farming systems. Milk from extensive farms exhibited higher MDE concentrations compared to milk from intensive farms. Additionally, the presence of miRNA *bta-miR-451* in milk exosomes was significantly elevated in grazing farms compared to that in intensive farming, which may be beneficial given its involvement in suppressing cell proliferation in different types of cancer. Our findings highlight the influence of the dairy production system on the functional properties of milk and may help guide future research on how miRNAs contribute to the functionality of milk as well as other agricultural food products.

## Figures and Tables

**Figure 1 animals-14-02960-f001:**
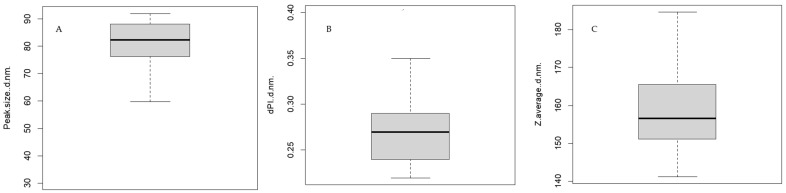
Physical characteristics of extracellular vesicles from milk produced on eight extensive or nine intensive dairy farms, based on DLS. Results for the 17 samples are summarised in Tukey plots. (**A**) Diameter (nm) of the most abundant peak in each sample. (**B**) Z-value of the diameters of the most abundant peak in each sample. (**C**) Distribution of polydispersity index in each sample.

**Figure 2 animals-14-02960-f002:**
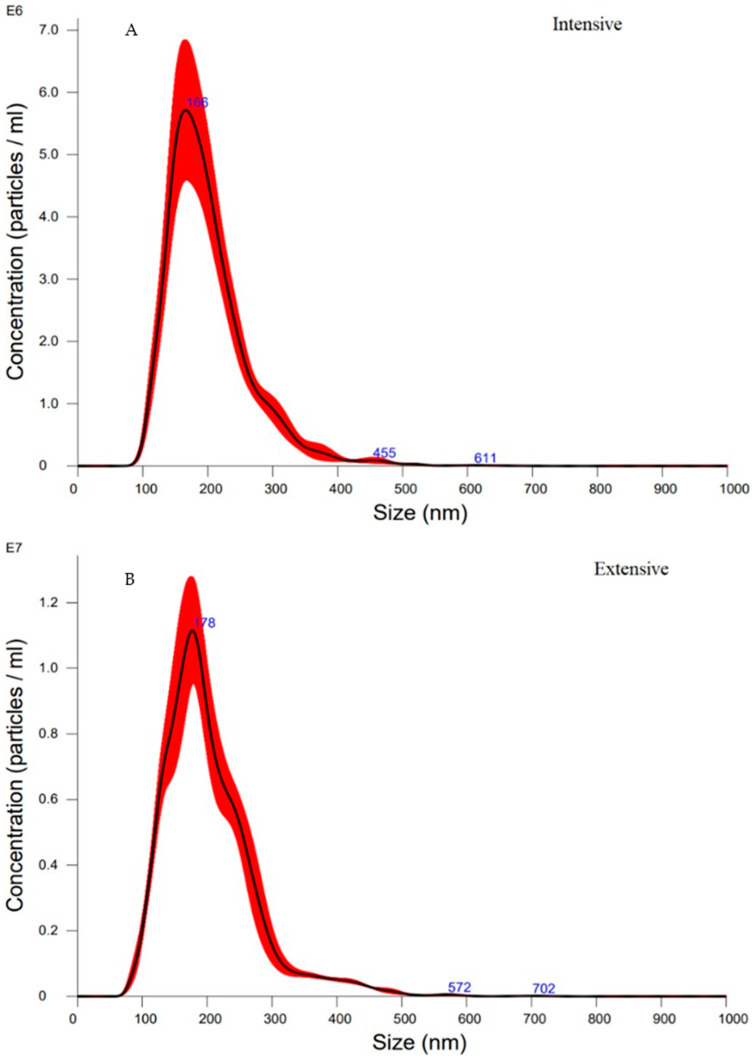
Average size distribution of MDEVs from (**A**) nine intensive dairy farms and (**B**) eight extensive dairy farms.

**Figure 3 animals-14-02960-f003:**
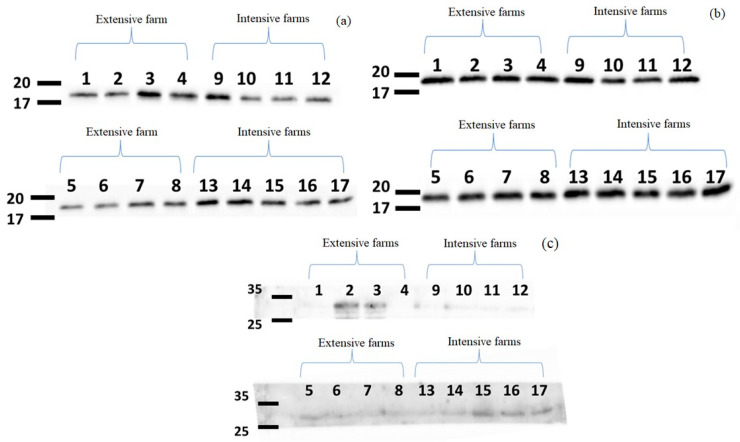
Representative Western blots of proteins in MDEVs from milk produced on eight extensive or nine intensive dairy farms. The fractionated vesicle proteins were immunoblotted against the exosome markers (**a**) CD 9 and (**b**) CD 81, as well as (**c**) the micelle marker casein. Molecular weight markers (in kDa) are depicted at the left of each gel. The lanes are numbered according to the Farm IDs in Table 1.

**Figure 4 animals-14-02960-f004:**
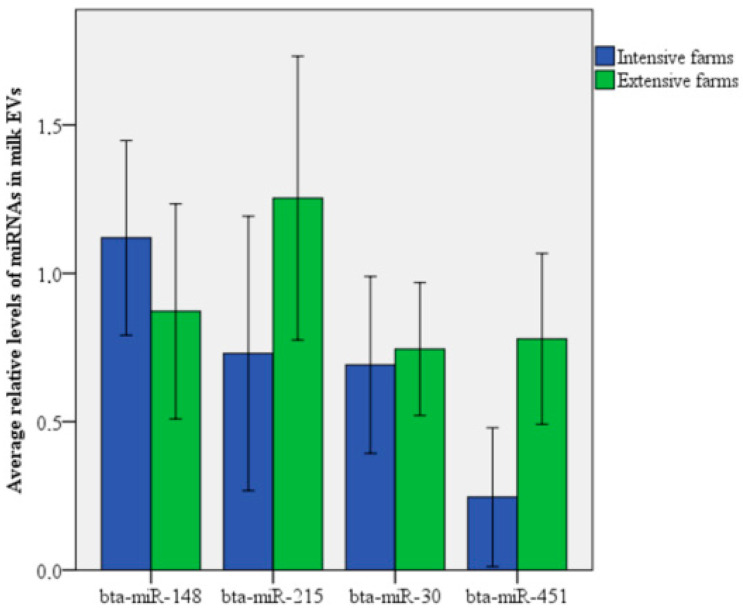
Relative levels of four miRNAs in MDEVs in milk produced on nine intensive dairy farms (blue) and eight extensive dairy farms (green). *p* < 0.05.

**Table 1 animals-14-02960-t001:** Characteristics of dairy farms where milk was sampled in this study.

Characteristic	Farm ID	Number of Cows	Average Daily Milk Production per Cow, L	Grazing	Daily Corn Silage per Cow, kg Fresh Matter	Daily Grass Silage per Cow, kg Fresh Matter	Daily Straw per Cow, kg Fresh Matter	Daily Vetch per Cow, kg Fresh Matter	Daily Alfalfa per Cow, kg Fresh Matter	Daily Concentrate per Cow, kg Fresh Matter
Extensive	1	41	21	Yes	0	5	0	3	0	6
	2	38	23	Yes	0	0	0	1	1	7
	3	35	15	Yes	0	0	0	0	0	4.5
	4	61	25	Yes	0	5	0	0	0	10
	5	14	25	Yes	0	5	0	0	0	9
	6	30	20	Yes	0	0	0	0	4	7
	7	35	26	Yes	0	5	0	2	2.5	9
	8	24	23	Yes	0	0	0	0	2	8
Intensive	9	240	36	No	16	10	0.5	2	2	12
	10	60	32	No	20	20	0	1.5	0	10
	11	75	34	No	25	8	0	0	2	11.5
	12	50	30	No	20	6	1	0	2	7
	13	44	34	No	20	10	0	1.5	0	10
	14	66	35	No	25	10	0	0	0	12
	15	120	36	No	30	5	1	2	1.5	12
	16	85	34	No	22	15	1	1.5	2	11
	17	66	40	No	25	10	0	0	2	12

**Table 2 animals-14-02960-t002:** Pathways in the Kyoto Encyclopedia of Genes and Genomes that are associated with the genes targeted by the human miRNA *miR-451*.

Pathway	*p*	Number of Genes Targeted by *miR-451*
Parkinson’s disease	<0.01	2
mTOR signalling pathway	<0.01	5
Thyroid cancer	<0.01	2
Colorectal cancer	0.03	3

## Data Availability

The data that support the findings of this study are available from the corresponding author upon reasonable request.

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
