# Peer review of "Association Between Dairy Production System and Milk Functionality Based on Analysis of miRNAs in Exosomes from Milk"

_animals, 2024, doi:10.3390/ani14202960_

Round 1

Reviewer 1 Report

Comments and Suggestions for Authors

This study is very interesting especially for the possible correlation with some degenerative diseases in humans. The study highlights the difference, in terms of the presence of bioactive molecules, in the milk of animals raised extensively compared to animals raised intensively. They therefore highlight how farming practices significantly influence the nutritional and functional properties of milk. This study compares miRNA profiles in milk exosomes from 19 extensive (pasture-based) and intensive (confined housing) dairy farms and explores their potential implications for human consumers. Based on the results, the authors conclude by underlining that extensive systems produce higher quality milk with better animal welfare compared to intensive systems.

Reviewer 2 Report

Comments and Suggestions for Authors

Exosomes are secreted vesicles which can transmit molecular cargo between cells. Exosomal microRNAs (exomiRs) have drawn much attention in recent years because there is increasing evidence to suggest that loading of microRNAs into exosomes is not a random process. Exosomes are continuously released and recycled by cells. Through the process of endocytosis, exosomes re-enter cells, where they are called endosomes. Endosomes are packaged together in multivesicular bodies (MVBs). MVBs rich in cholesterol are trafficked to the cell membrane where they fuse and are released as exosomes, whilst those which are cholesterol deficient are recycled through lysosomes. Exosomes (EXOs) are natural nanoparticles of endosome origin that are secreted by a variety of cells in the body. Exosomes have been found in bio-fluids such as urine, saliva, amniotic fluid, and ascites, among others. Milk is the only commercially available biological liquid containing EXOs. Proof that exosomes are essential for cell-to-cell communication is increasingly being reported. Studies have shown that they migrate from the cell of origin to various bioactive substances, including membrane receptors, proteins, mRNAs, microRNAs, and organelles, or they can stimulate target cells directly through interactions with receptors. Because of the presence of specific proteins, lipids, and RNAs, exosomes act in physiological and pathological conditions in vivo.

Based on all of the above, we conclude that this research is valid, and the title is adequate because EXOs have been very significant lately.

The present research compares milk from extensive (pasture-based) and intensive (confined housing) dairy farms to see how different farming methods affect miRNA levels in milk exosomes. The authors discovered that milk from large farms had more extracellular vesicles. More precisely, the miRNA bta-miR-451 was discovered at higher levels in milk from large farms. The authors state that miRNA is identical to a human miRNA associated with major health pathways (Parkinson's disease and cancer).

Introduction is well written. Authors described intensive and extensive milk production and influence of type of production on microRNAs (miRNA). Then the authors state the importance of extracellular miRNA that can be found in the vesicular formations of milk. Using modern software, databases and bioinformatics, the authors examine miRNAs in the vesicular formations of milk originating from cows from extensive and intensive farming in order to determine their functional connection with processes in humans as the main milk consumer.

Line 25: explain EV

MandM are well described and are sufficient to reproduce the research. Please indicate the health status of the cows whose milk was used in the trial.

The results are clearly presented. It would be good to graphically present the pathway map so that readers can better understand the importance of the examined parameter and its potential importance for human health. I think there are variables whose influence could have been examined in this study (see later voltages).

A discussion follows the obtained results. The conclusion follows from the obtained results. The discussion is too narrow and oriented only on one main result, without a wider digression.

Limitations and additional questions:

This study has several limitations and unanswered questions. The main major limitation is that individual milk samples were not examined, farm effect or season effect and other significant parameters that could cause variability of miRNAs in exosomes from milk were not examined. There may be cows with certain traits that are superior in the process of obtaining a higher miRNA levels in milk exosomes.

In this experiment, the influence of the chemical composition of the meal on miRNA levels in milk exosomes was not examined. The chemical components of the meal can greatly affect the characteristics of the milk and everything in the milk.

I suggest you study the pre-analytical factors concerning milk sampling.

Also, several questions could be asked here that can be searched for an answer: 1) can the amount of milk produced affect miRNAs in exosomes from milk? 2) can the age of cows affect miRNAs in exosomes from milk? 3) iIs intrafarm variability within trial subgroups or within trials sufficient to affect miRNAs in exosomes from milk? 4) is the addition of a certain amount of the concentrated portion of the meal sufficient to affect miRNAs in exosomes from milk?.

You can answer certain questions by supplementing your research and expanding your manuscript. However, since the concept of your work is clear to me, I suggest that the editor of the journal determines whether the examination and presentation of the influence of certain variables on miRNA levels in milk exosomes is necessary for this work. In any case, you must answer most of the questions and enrich the discussion. A separate paragraph should be devoted to the limitations and potential future impact of this study, where the authors would raise and/or provide answers to the questions we discussed in the previous paragraphs.

Reviewer 3 Report

Comments and Suggestions for Authors

1. The presented article "Association between dairy production system and milk functionality based on analysis of miRNAs in exosomes from milk" deals with an issue that is interesting and important from the point of view of human nutrition. The article is systematically and clearly arranged, it is based on very well-founded research, but I still have a few comments and questions about it, which in no way diminish the very good quality of the research carried out.

2. I recommend slightly modifying the abstract and stating a clear conclusion as to whether the authors recommend, on the basis of their research, the breeding of dairy cows in intensive or extensive farms, or another form of clarification in accordance with the extended content of Conclusions.

3. The introduction shows the current development in this field of scientific research, which has been documented by 23 scientific publications.

4. Materials and Methods are clearly processed. Table 1 of the article shows the basic characteristics of the feed in more detail. In terms of milk quality, it would also be advisable to specify the method of milking and milk treatment (milking technology in milking parlours, AMS with milking robots, mobile milking on pasture, etc.), especially on farms with extensive breeding of dairy cows.

5. In the Results section, a summary overview of the obtained research results is systematically presented.

6. In the Discussion, I would expect a more significant analysis of the causes of better or worse conditions and quality of milk. The authors should indicate whether the causes of the differences are only in the quality of the feed, or whether the influence of different housing methods, milking techniques and treatment of milk after milking (filtration and cooling), the influence of climatic conditions (e.g. heat stress) or other reasons?

7. Conclusions – are very brief and are based on the established research results. I wonder if, according to the authors, milk from pasture-raised farms is of better quality. The authors have made some research progress and difference compared to the article Abou el qassim, L.; Alonso, J.; Zhao, K.; Guillou, S. Le; Diez, J.; Vicente, F. Differences in the MicroRNAs Levels of Raw Milk 365 from Dairy Cattle Raised under Extensive or Intensive Production Systems. 2022, 1–16. But this older paper has the same conclusion, briefly summarized: "the authors found that there is a difference in milk from extensive and intensive farms". Unfortunately, they do not indicate the causes or ways of possible improvement.

8. The conducted research gives a good potential for greater progress in the solution and contribution of the conducted research than just stating that there is a difference between the two methods of farm management, which is reflected in the quality of the milk. Perhaps it would be interesting to prepare an analysis of the causes of the different quality based on further analysis of the input data on the farms and look for ways to achieve a balanced result in milk quality in both types of farms. I ask the authors for their opinion and explanation.

Round 2

Reviewer 2 Report

Comments and Suggestions for Authors

Thank you for your response. I wish you all the best in next research.